# Synthesis and Characterization of MnIn_2_S_4_/Single-Walled Carbon Nanotube Composites as an Anode Material for Lithium-Ion Batteries

**DOI:** 10.3390/nano14080716

**Published:** 2024-04-19

**Authors:** Pei-Jun Wu, Chia-Hung Huang, Chien-Te Hsieh, Wei-Ren Liu

**Affiliations:** 1Department of Chemical Engineering, R&D Center for Membrane Technology, Center for Circular Economy, Chung Yuan Christian University, 200 Chung Pei Road, Chungli District, Taoyuan City 320, Taiwan; bigbang71602@gmail.com; 2Department of Electrical Engineering, National University of Tainan, No. 33, Sec. 2, Shulin St., West Central District, Tainan City 700, Taiwan; chiahung@mail.mirdc.org.tw; 3Metal Industries Research and Development Centre, Kaohsiung 701, Taiwan; 4Department of Chemical Engineering and Materials Science, Yuan Ze University, Taoyuan 320, Taiwan; 5Department of Mechanical, Aerospace, and Biomedical Engineering, University of Tennessee, Knoxville, TN 37996, USA

**Keywords:** lithium-ion battery, transition metal sulfides, MnIn_2_S_4_, single-walled carbon nanotubes, anode

## Abstract

In this study, we synthesized a transition metal sulfide (TMS) with a spinel structure, i.e., MnIn_2_S_4_ (MIS), using a two-step hydrothermal and sintering process. In the context of lithium-ion battery (LIB) applications, ternary TMSs are being considered as interesting options for anode materials. This consideration arises from their notable attributes, including high theoretical capacity, excellent cycle stability, and cost-effectiveness. However, dramatic volume changes result in the electrochemical performance being severely limited, so we introduced single-walled carbon nanotubes (SWCNTs) and prepared an MIS/SWCNT composite to enhance the structural stability and electronic conductivity. The synthesized MIS/SWCNT composite exhibits better cycle performance than bare MIS. Undergoing 100 cycles, MIS only yields a reversible capacity of 117 mAh/g at 0.1 A/g. However, the MIS/SWCNT composite exhibits a reversible capacity as high as 536 mAh/g after 100 cycles. Moreover, the MIS/SWCNT composite shows a better rate capability. The current density increases with cycling, and the SWCNT composite exhibits high reversible capacities of 232 and 102 mAh/g at 2 A/g and 5 A/g, respectively. Under the same conditions, pristine MIS can only deliver reversible capacities of 21 and 4 mAh/g. The results indicate that MIS/SWCNT composites are promising anode materials for LIBs.

## 1. Introduction

Rapid technological breakthroughs have increased the demand for energy storage devices with a high energy density and quick charging and discharging capabilities. Among these, LIBs have garnered the most attention because of their elevated energy density, sustained operating voltage, and extended cycle lifespan [1,2]. Currently, the anode material commonly used in commercial LIBs is graphite [3,4]. However, its low theoretical capacity and power density restrict its usage in several applications. Therefore, it is imperative to find an alternative to graphite. LIBs can be classified into three categories of anode materials according to their ion storage mechanisms: intercalation materials, conversion materials, and alloy materials. Notably, conversion materials, predominantly comprising transition metal compounds, have emerged as a research focus in recent times. Transition metal oxides (TMOs) are less expensive and more accessible than graphite. However, they still encounter issues such as significant volume expansion and poor conductivity.

Many researchers have turned their focus to TMSs, with the spinel structure proving particularly attractive. Numerous binary TMSs, including CoS_2_ [5], CuS [6], In_2_S_3_ [7], MoS_2_ [8], and ZnS [9], have been proposed as suitable LIB anode materials. However, compared to binary TMSs, with AB_2_S_4_, ternary TMSs (A and B are both transition metals, but they are not the same element; the valence of A is +2, and for B it is +3) exhibited superior electronic conductivity and redox activity, resulting in better capacity performance [10]. In recent years, ternary TMSs such as MnCo_2_S_4_ [11], NiCo_2_S_4_ [12,13,14,15], and CuCo_2_S_4_ [16,17,18] have received significant attention. Kim et al. developed NiTi_2_S_4_ and TiS_2_ as anode materials for LIBs. In their study, NiTi_2_S_4_ electrodes had higher reversible capacities of 820, 759, 700, 651, and 570 mAh/g at current densities of 100, 200, 500, 1000, and 2000 mA/g. Conversely, as the current density increased, the specific capacities of the TiS_2_ (588, 258, 116, 47, and 15 mAh/g) electrodes rapidly dropped [19]. To this end, many researchers have explored the use of carbon-based composites, such as carbon nanotubes (CNTs) [10,20,21,22], graphene composites [23,24,25,26,27,28], carbon nanofibers (CNFs) [7,29], and carbon black [30] to mitigate the structural damage brought on by volume expansion and enhance the conductivity of ternary TMSs. Lin et al. proposed a NiCo_2_S_4_/CNT nanocomposite using a rapid and straightforward hydrothermal process. NiCo_2_S_4_/CNT demonstrated significant reversible capacities of 1388, 1257, 1194, 1172, and 1022 mAh/g with corresponding current densities of 0.2, 0.5, 0.8, 1, and 2 C. Notably, upon returning to a current density of 0.2 C, NiCo_2_S_4_/CNT maintained a high capacity of 1208 mAh/g [31]. Muralidharan and Nallathamby reported CuCo_2_S_4_/multi-walled carbon nanotube (MWCNT) nanoparticles synthesized through a two-step solvothermal and sintering method. The internal resistance values of 35 Ω and 194 Ω of the CuCo_2_S_4_ anode were found to reduce to 15 Ω and 79 Ω in the composite, respectively [32]. This suggests that incorporating MWCNTs can significantly enhance the transfer of interface electrons in the material. The reversible capacities of the CuCo_2_S_4_/MWCNT electrode were 1374, 1270, 1094, 920, and 735 mAh/g for current densities of 50, 100, 200, 500, and 1000 mA/g, respectively. Remarkably, after reverting to a current density of 100 mA/g, CuCo_2_S_4_/MWCNTs illustrated a high reversible capacity of 1207 mAh/g. The incorporation of CNTs into anode materials has been proven to be effective in alleviating the significant structure expansion that occurs during charge and discharge cycles. CNT additives enhance the cycle stability and bolster the electronic conductivity. Manganese can be used as a low-cost additive, and studies have also been published on indium-related sulfides [23,29]. However, indium is considered a relatively rare and expensive metal because of its limited reserves, high demand with respect to electronics, and recycling challenges. The future price of indium might depend on various factors, including changes in supply and demand, technological developments, and exploration of new sources. While predicting specific outcomes is challenging, there are some potential outcomes that could contribute to a reduction in the price of indium. Thus, it is necessary to study the electrochemical behavior of MIS/SWCNT composites as an anode for LIBs.

In this study, we used a two-step hydrothermal and sintering approach to synthesize pristine MIS and an MIS/SWCNT composite. The crystal structure, surface morphology, elemental distribution, specific surface area, pore size distribution, charge distribution, and carbon content were evaluated using a range of techniques. These included X-ray diffraction (XRD) for crystal structure analysis, scanning electron microscopy (SEM) for surface morphology observation, high-resolution transmission electron microscopy (HRTEM) for detailed imaging, energy-dispersive X-ray spectroscopy (EDX) for elemental distribution analysis, the Brunauer–Emmett–Teller method (BET) for assessing surface area, X-ray photoelectron spectroscopy (XPS) for surface chemical composition analysis, and thermogravimetric analysis (TGA) for studying carbon content changes. Finally, we assembled half cells and investigated their electrochemical storage performance and electrochemical behavior. The MIS modified by SWCNTs showed a good rate performance and excellent cycle stability, and its electrochemical performance was also significantly improved compared to pristine MIS.

## 2. Materials and Methods

### 2.1. Synthesis of Pristine MIS and MIS/SWCNT Composites

The MIS/SWCNT composite was synthesized via a two-step hydrothermal sintering method. Typically, 1 wt.% of SWCNTs (Sino Applied Technology, Taoyuan, Taiwan) was dispersed in 50 mL of deionized (DI) water for a duration of 30 min. Following this, a solution containing 1 mmol of manganese nitrate tetrahydrate (Mn(NO_3_)_2_·4H_2_O, 98% purity, Alfa-Aesar, Ward Hill, MA, USA), 2 mmol of indium nitrate (In(NO_3_)_3_, 99.99% purity, Alfa-Aesar), 8 mmol of thioacetamide (CH_3_CSNH_2_, 99% purity, Alfa-Aesar), and 15 mmol of urea (CH_4_N_2_O, 99% purity, Sigma-Aldrich, St. Louis, MO, USA) was introduced into the aforementioned dispersion. Subsequently, the mixture underwent vigorous stirring for a duration of two hours, ensuring its uniformity. After being put into a 100 mL Teflon-lined stainless-steel autoclave, the homogenous solution was allowed to react for 24 h at 180 °C. Subsequently, the autoclave was left to cool naturally to ambient temperature. Next, the obtained powder was collected via high-speed centrifugation and extensively rinsed multiple times with DI water and ethanol. Eventually, the powder was dried for 12 h in a vacuum oven set to 60 °C. Subsequently, the powder underwent an additional annealing process, being heated to 350 °C at a rate of 5 °C per minute for a duration of 2 h in an argon environment. The aim of this step was to enhance crystallinity and generate the MIS/SWCNT composite. In contrast, the creation of pristine MIS involved a comparable process but the addition of SWCNTs was skipped. A flowchart of MIS/SWCNT composite creation is shown in Figure 1.

### 2.2. Characterizations

The crystal structure and phase purity of synthesized samples were analyzed using an X-ray diffractometer (Bruker, D2-Phaser, Billerica, MA, USA) with Cu-Kα radiation at a scan rate of 5°/min and a scanning angle range from 10° to 80°. The surface morphology of the samples was examined using a field-emission scanning electron microscope (JEOL JSM-7600F, Zhubei City, Taiwan) and a high-resolution transmission electron microscope (JEOL JEM2100). The specific surface area and pore size distribution were determined via the BET method by employing a Micromeritics Tristar 3000 (Norcross, GA, USA). Additionally, the valence state of compounds was examined using an X-ray photoelectron spectrometer (Thermo Fisher Scientific, Waltham, MA, USA), which uses X-rays as the detection source.

### 2.3. Electrochemical Measurements

The active materials (MIS or MIS/SWCNTs), conductive additive (Super-P carbon, Timcal^®^, Shanghai, China), and organic-based binder polyvinylidene were dissolved in N-methylpyrrolidone at a weight ratio of 8:1:1. Subsequently, the resulting slurry was applied to a copper foil to create the working electrode. Next, the electrodes were introduced into a vacuum oven set at 120 °C for a duration of 9 h. Then, a CR2032 coin-type half-cell was built in a glove box filled with argon. The LIB electrolyte utilized in the experiment consisted of a 1 M LiPF_6_ solution diluted at a weight ratio of 1:1:1 with a combination of ethylene carbonate, ethyl methyl carbonate, and dimethyl carbonate. The potential range of the coin-type half-cell (CR2032) during charging and discharging under various current densities is 0.01~3 V. Furthermore, the frequency range of the electrochemical impedance spectrum AC impedance setting is 10 mHz to 1 MHz.

## 3. Results and Discussion

The XRD patterns of pristine MIS and MIS/SWCNT composites are illustrated in Figure 1a. All of the diffraction peaks match the ICSD standard pattern (ICSD-639973). The results indicate that both MIS and the MIS/SWCNT composite were pure. Appendix A displays the grain size of pristine MIS and the as-synthesized MIS/SWCNT composite calculated using the Scherrer equation. Compared with pristine MIS, it can be observed that the grain size of the MIS/SWCNT composite has obviously decreased, which also means that the use of SWCNTs for modification will inhibit the agglomeration and nucleation of MIS. The crystal structure of MIS is illustrated in Figure 1b. MIS has a face-centered cubic spinel structure with an *Fd3̅m* space group, in which Mn^2+^ occupies tetrahedron 8a sites, In^3+^ occupies octahedron 16d sites, and S^2−^ occupies tetrahedron 32e sites, forming a corner-sharing tetrahedral network.

In order to analyze the surface morphology and structure of MIS, the MIS/SWCNT composite, and SWCNTs, field-emission scanning electron microscopy (FE-SEM) was used. The SEM image of unmodified MIS is depicted in Figure 2a. The SEM results indicate that pristine MIS has irregular aggregate particles that are around 1–2 mm in size. By comparison, as illustrated in Figure 2b, MIS/SWCNT composite particles exhibit a small tubular structure on the surface. The SEM image of SWCNTs is displayed in Figure 2c. These results demonstrate that MIS is successfully wrapped by the SWCNTs. The EDX mapping of pristine MIS is presented in Figure 2d. The molar concentrations of Mn, In, and S in MIS were determined to be 11, 59, and 30%, respectively.

To confirm the carbon content of the MIS/SWCNT composite, we conducted a thermogravimetric analysis with a heating rate of 10 °C per minute from ambient temperature to 800 °C in an air atmosphere. As displayed in Appendix A, between 35 °C and 180 °C, MIS and the MIS/SWCNT composite showed a slight weight loss that was caused by evaporation of the water in the sample [7]. Upon reaching a temperature of 350 °C, a slight increase in weight was observed. This phenomenon can be attributed to the oxidation of sulfides, leading to the formation of metal oxides [33]. With the temperature was elevated to 550 °C, a gradual decrease in weight was observed. This phenomenon can be attributed to the decomposition of metal oxides present in the sample [34]. This trend continued until approximately 750 °C. Compared to the TGA curves of pristine MIS, it can be concluded that the carbon content of the MIS/SWCNT composite was about 10.07 wt.%. It was speculated that, during the centrifugation process, a small amount of metal MIS was still dissolved in the DI water and ethanol and was not precipitated, which caused a discrepancy between the amount of SWCNTs added and the actual loading amount.

Further insight into the intricate nanostructure of both the pristine MIS and the MIS/SWCNT composite was gained through transmission electron microscopy (TEM) analysis. Figure 3a illustrates the agglomerated particles of MIS, with an average particle size measuring around 1–2 μm. The MIS HRTEM image and the accompanying selected-area electron diffraction (SAED) pattern are displayed in Figure 3b. The analysis shows that lattice spacings of 0.189 nm and 0.204 nm correspond to the (440) and (333) crystallographic planes, respectively. The SAED pattern displayed in Figure 3b also confirms that MIS has a polycrystalline nature. The TEM images of the MIS/SWCNT composite are shown in Figure 3c,d. It can be observed that some SWCNTs exist on the MIS surface. In order to see the tubular structure of SWCNTs more clearly, Appendix A demonstrates the HRTEM image of SWCNTs. And the diameter of SWCNTs was observed ~10 nm. The HRTEM images and SAED patterns of the MIS/SWCNT composite are displayed in Figure 3e. The (311), (400), (511), and (620) planes for the different rings are similar to the SAED pattern that was obtained for the MIS/SWCNT composite. The EDX elemental mapping from the TEM image of the MIS/SWCNT composite is shown in Figure 3f. Mn, In, S, and C are uniformly distributed throughout the sample, which is consistent with the SEM-EDX mapping results in Figure 2d.

The nitrogen adsorption/desorption isotherms and pore size distributions for the virgin MIS and the MIS/SWCNT composite are presented in Figure 4a–d. Both samples exhibited type-IV curves with hysteresis loops, indicating that the material had a mesoporous structure. The specific surface area curves of pure MIS and the MIS/SWCNT composite are shown in Figure 4a,c. The pore size distributions of pristine MIS and the MIS/SWCNT composite determined via the Barrett–Joyner–Halenda model are depicted in Figure 4b,d. The specific surface areas of the pristine MIS and the MIS/SWCNT composite were 50.16 m^2^/g and 39.77 m^2^/g, respectively. The average pore diameters of MIS and the MIS/SWCNT composite were 9.00 nm and 10.67 nm, respectively. The surface area was drastically reduced after compositing with SWCNTs, but the average pore size rose noticeably. This could be attributed to the fact that SWCNTs cover the smaller micropores on MIS, resulting in a reduced contact area with nitrogen [23,35].

XPS was used to analyze the MIS/SWCNT composites’ elemental make-up and chemical bonding states. The XPS spectrum for the MIS/SWCNT composite is depicted in Appendix A. This spectrum confirms the presence of elements such as Mn, In, S, and C, which correlate with the elemental distribution observed in the EDX mapping spectrum shown in Figure 3f. The two deconvoluted peaks in Figure 5a may be attributed to the contributions of Mn 2p_3/2_ and Mn 2p_1/2_, and are present at 641.8 eV and 653.2 eV, respectively. Furthermore, the satellite peak at 645.4 eV is a result of the multiple splitting of Mn ion energy levels. Two peaks at 444.9 eV and 452.5 eV, which represent the In 3d_5/2_ and In 3d_3/2_ states, respectively, can also be seen in the In 3d spectrum in Figure 5b. These results are in accordance with the three valence states of In in MIS. The spectrum of the S 2p region is displayed in Figure 5c. This spectrum can be separated into three main peaks. The peaks observed at roughly 161.5 eV and 162.7 eV can be attributed to S 2p_3/2_ and S 2p_1/2_, respectively, whereas the peak at 162.6 eV can be attributed to a metal–sulfur (M-S, M=Mn or In) bond. The deconvoluted C 1s spectra, as shown in Figure 5d, have two primary peaks located at around 284.9 eV and 286.0 eV, which were assigned to C=C/C-C and C-S, respectively.

To determine the electrochemical reactions occurring during the charge and discharge process of the MIS and MIS/SWCNT anode electrodes, we conducted cyclic voltammetry (CV) measurements. Figure 6a,b show the CV curves of MIS and MIS/SWCNT composite during the first three cycles, with a scan rate of 0.1 mV/s, in the potential range 0.01–3.0 V. During the initial cycle, the reduction peaks observed in the voltage range of 1.3–1.6 V were attributed to the conversion of MIS into MnS [36]. This reduction resulted in cathodic peaks of 0.92 V and 0.26 V due to the formation of in-metallic and stable solid electrolyte interphase (SEI) layers on the MIS electrode [7]. The anodic peak at 0.41 V in the subsequent cycles illustrates the process of reducing MnS to metallic Mn [37]. The potential peak observed at 0–0.1 V is indicative of the insertion/intercalation of Li^+^ ions into the carbon structure of the electrode [38]. The oxidation peaks detected at 0.43 V and 0.68 V are attributed to the Mn-alloy delithiation process, whereas the anodic peak at 1.14 V indicates Li_x_In delithiation [7,26]. The peaks in the 1.4–2.4 V potential range are related to the production of Mn-sulfide and In-sulfide molecules [33]. The SWCNT-modified MIS electrode exhibited similar redox peaks to pristine MIS. Additionally, MIS/SWCNT composite exhibited better overlapping than pristine MIS. This also means that the MIS/SWCNT electrode demonstrated excellent electrochemical performance. Based on the CV test results, the electrochemical reaction process of MIS was determined as follows:*MnIn*_2_*S*_4_ + 6*Li*^+^ + 6*e*^−^ → *MnS* + 2*In* + 3*Li*_2_*S*
(1)
*MnS* + 2*Li*^+^ + 2*e*^−^ ↔ *Mn* + *Li*_2_*S*(2)
*In* + *xLi*^+^ + *xe*^−^ ↔ *Li*_*x*_*In*
(3)
or
*MnIn*_2_*S*_4_ + 8*Li*^+^ + 8*e*^−^ → *Mn* + 2*In* + 4*Li*_2_*S*
(4)
*Mn* + *xLi*^+^ + *xe*^−^ ↔ *Li*_*x*_*Mn*
(5)
*In* + *xLi*^+^ + *xe*^−^ ↔ *Li*_*x*_*In*
(6)

Figure 6c,d display pristine MIS and MIS/SWCNT electrode galvanostatic discharge–charge curves at 0.1 A/g for the first three cycles, with a potential window of 0.01–3.0 V. It was found that, during the first cycle, pristine MIS had an initial Coulombic efficiency of 51.1% with discharge and charge capacities of 1314 mAh/g and 671 mAh/g, respectively. A more irreversible capacity loss was brought on by the formation of SEI layers on the electrode surface and the suppression of Li^+^ insertion [11]. Subsequently, during the second/third cycles, the discharge and charge capacities were 722/542 and 527/444 mAh/g with Coulombic efficiencies of 70.2% and 84.3%, respectively. At the same current density, the initial discharge and charge capacities for MIS/SWCNT electrodes were 1249 mAh/g and 771 mAh/g, respectively, with a primary Coulombic efficiency of 61.7%. The reversible capacities for the second and third cycles were 780/702 and 717/675 mAh/g, with improvements in the Coulombic efficiencies of 90.0% and 94.1%, respectively. Additionally, when compared to pristine MIS, the galvanostatic charge–discharge curves of the MIS/SWCNT composite display a substantial overlap. This overlapping suggests that the synthesized MIS/SWCNT composite possesses excellent stability throughout the charge and discharge processes.

Figure 7a displays the rate capability of pure MIS and the MIS/SWCNT composite at various current densities. The pristine MIS discharge capacities were 525, 260, 133, 66, 21, and 4 mAh/g at current densities of 0.1, 0.2, 0.5, 1, 2, and 5 A/g, respectively. An average reversible capacity of only 234 mAh/g was attained when the current density reverted to 0.1 A/g. Conversely, the synthesized MIS/SWCNT composite demonstrated superior reversible capacities of 706, 532, 413, 324, 232, and 102 mAh/g in comparison to those of the bare MIS at the same current densities. Even with a reduced current density of 0.1 A/g, the MIS/SWCNT composite maintained a higher capacity of 523 mAh/g. The cycling performance of pristine MIS and MIS/SWCNT composite is illustrated in Figure 7b. A reversible capacity of only 117 mAh/g was maintained by pristine MIS after 100 cycles with a 0.2 A/g current density. Its capacity decayed rapidly over these cycles. In contrast, the MIS/SWCNT composite retained a high reversible capacity of 536 mAh/g at a current density of 0.2 A/g. Figure 7c–f depicts the corresponding galvanostatic discharge–charge curves for pristine MIS and the MIS/SWCNT composite for different current densities and cycles. The results reveal that SWCNT-modified MIS demonstrates enhanced overlapping behavior, contributing to improved battery stability and capacity performance. This observation indirectly substantiates the efficacy of utilizing SWCNTs as a means of enhancing anode materials.

To explore the conductivity and kinetic characteristics of pristine MIS and MIS/SWCNT composite, electrochemical impedance spectroscopy (EIS) plots were employed. Figure 8a illustrates the EIS plots for the initial MIS and the MIS/SWCNT composite after 2.5 cycles. The equivalent circuit diagram is shown in the inner figure in Figure 8a. The fitted data and calculated diffusion coefficients are shown in Appendix A. In the area of intermediate frequency, R_CT_ denotes the charge transfer resistance, R_SEI_ denotes the resistance of the SEI film, and R_S_ denotes the electrolyte resistance at the highest frequency. The slope of the line at low frequencies indicates the substance’s diffusion coefficient. As shown in Figure 8a and Appendix A, the MIS/SWCNT electrode illustrated two smaller semicircles compared to pristine MIS. These results indicate that the resistance of the MIS/SWCNT composite was significantly lower than that of pristine MIS. The linear connection between Z’ and ω^−1/2^ in the low-frequency domain is shown in Figure 8b. The formula below was used to calculate the diffusion coefficient of lithium ions (D_Li+_).
(7)$$D=\frac{R^{2}T^{2}}{2A^{2}n^{2}F^{4}C^{2}\sigma^{2}}$$

In Equation (7), the variables represent the following: *R* signifies the ideal gas constant, *T* stands for the ambient temperature, *A* denotes the electrode contact area, *n* signifies the number of electrons engaged during this reaction, *F* represents the Faraday constant, *C* denotes the concentration of Li^+^, and *σ* symbolizes the slope obtained from the fitted diffusion coefficient line. Based on Equation (7), the diffusion coefficient of MIS/SWCNT composite was measured to be 1.7 × 10^−13^ cm^2^/s, which is higher than the value of 1.7 × 10^−14^ cm^2^/s observed for pristine MIS. These EIS results suggest that modification with SWCNTs can indeed enhance the electrical conductivity, consequently leading to improved electrochemical performance.

To investigate the structural and morphological alterations that these materials undergo during the charging and discharging processes, we observed the SEM images of pristine MIS and MIS/SWCNT electrodes after 100 cycles, which are displayed in Figure 9a–d. Initially, there is no obvious difference between the MIS/SWCNT composite and pristine MIS before charging and discharging. However, after 100 cycles, it is apparent that there are numerous cracks and pores on the pristine MIS electrode, and there is obvious aggregation. Conversely, the MIS/SWCNT electrode is protected by SWCNTs, which inhibits the dramatic volume expansion and contraction resulting from lithium ion insertion and deintercalation. As a result, the morphology of the MIS/SWCNT electrode is maintained, without any fractures or damage. Via the introduction of SWCNTs into MIS, not only is the volume expansion of MIS buffered, but the electronic conductivity of the MIS anode is significantly enhanced, which further improves its electrochemical performance. Appendix A and Appendix A further compare the rate performance and cycle life of MIS/SWCNT composite and other reported TMS anode materials for LIBs [7,25,29,39,40,41]. Overall, the MIS/SWCNT composite exhibits better electrochemical performance than other TMSs. SWCNTs play a crucial role in providing good protection to the electrode’s structure, which helps prevent crack formation and aggregation. Additionally, SWCNTs improve the electronic conductivity of MIS during the process of Li^+^ insertion and extraction, resulting in an overall improvement in battery performance. Due to these synergistic effects, SWCNTs are recognized as a valuable component for the modification of anode materials, improving the performance and stability of LIBs.

## 4. Conclusions

As a result of hydrothermal synthesis and high-temperature sintering, a transition metal sulfide with a unique spinel structure (MnIn_2_S_4_) was effectively synthesized. To solve the problems of volume expansion and agglomeration, we introduced SWCNTs into MIS during the hydrothermal process. The rate capability results show that, at a current density of 0.1 A/g, the MIS/SWCNT composite initially had a high reversible capacity of 706 mAh/g. Even when the current density was increased to 5 A/g, the capacity remained at 102 mAh/g. Furthermore, the MIS/SWCNT composite maintained a specific capacity of 536 mAh/g even after undergoing 100 cycles with a current density of 0.2 A/g. In the EIS tests, it was found that the semicircle of the SWCNT-modified MIS was smaller than that of pristine MIS, and the diffusion coefficient was also significantly improved. The results obtained from the experiments clearly demonstrate that utilizing SWCNTs to modify MIS is a potential approach to improving the lithium storage performance and overall electrochemical performance. The enhanced stability, structural integrity, and improved conductivity of the MIS/SWCNT electrode contribute to better cycling stability and capacity retention during repeated charging and discharging cycles. The success of the MIS/SWCNT composite in these experiments suggests that it holds considerable potential for practical application in LIBs. The use of MIS/SWCNT electrodes is a promising avenue by which to enhance the energy storage capacity and efficiency of LIBs. This advancement holds the potential to contribute to the creation of more reliable and high-performance energy storage devices due to the heightened performance and stability MIS/SWCNT electrodes offer.

## Data Availability

Data are contained within the article.

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
