# Peer review of "Synthesis and Characterization of MnIn2S4/Single-Walled Carbon Nanotube Composites as an Anode Material for Lithium-Ion Batteries"

_nanomaterials, 2024, doi:10.3390/nano14080716_

Round 1
Reviewer 1 Report
Comments and Suggestions for Authors
Synthesis and characterizations of MnIn2S4/SWCNTs compo[1]site as anode materials for Lithium-ion batteries.
1. The introduction section is prolix and still could not highlight the major objectives and findings.
2. Extensive English correction is necessary. Some sentences are arranged improperly.
3. ECSA should be calculated.
4. The rational design of the transition metal sulfide can be highlighted in the introduction section with the reference of following articles:
doi.org/10.1016/j.compositesb.2022.110339
5. What changes can be seen in the electronic structure of MnIn2S4/SWCNTs after long term stability test.
Comments on the Quality of English LanguageExtensive English correction is necessary.
Author Response
Reviewer 1
Synthesis and characterizations of MnIn2S4/SWCNTs composite as anode materials for Lithium-ion batteries.
- The introduction section is prolix and still could not highlight the major objectives and findings.
Response:
Thank you for reviewer’s comments, we have highlighted the major objectives and findings in the revised introduction.
“Despite their superior properties, ternary transition metal sulfides still have certain limita-tions that need to be addressed and improved, such like structural damage caused by se-vere volume expansion and poor performance of ion transfer because of significant ag-glomeration.”
“Finally, the MIS modified by SWCNTs showed high-rate performance and excellent cycle stability, and its electrochemical performance was also significantly improved compared to pristine MIS.”
We hope reviewer could satisfy with our response.
- Extensive English correction is necessary. Some sentences are arranged improperly.
Response:
Thanks to the reviewer's comments. The manuscript has been sent to the English editing. Please see the corresponding certificate.
The certificate of English editing for our manuscript.
- ECSA should be calculate.
Response:
Thanks to the reviewer's comments, Because ESCA analyses only display the surface information of the testing sample. If we would like to identify the composition of our as-synthesized MnIn2S4 by ESCA quantitatively analyses may not a suitable tool. We hope reviewer could satisfy with our response.
- The rational design of the transition metal sulfide can be highlighted in the introduction section with the reference of following articles:
doi.org/10.1016/j.compositesb.2022.110339
Response:
Thanks to the reviewer's suggestions. In order to demonstrate the rational design of transition metal sulfide, we have cited the important reference in the revised manuscript.
- Poudel, M.B.; Kim, A.R.; Ramakrishan, S.; Logeshwaran, N.; Ramasamy, S.K.; Kim, H.J.; Yoo, D.J. Integrating the essence of metal organic framework-derived ZnCoTe–N–C/MoS2 cathode and ZnCo-NPS-N-CNT as anode for high-energy density hybrid supercapacitors. Composites Part B: Engineering 2022, 247, 110339.
- What changes can be seen in the electronic structure of MnIn2S4/SWCNTs after long term stability test.
Response:
Thanks to the reviewer's comments. We do believe there is no obvious change of electronic structure in MIS/SWCNTs after long term stability tests. First, we used cyclic voltammetry (Fig. 6a and 6b) to observe the redox reaction of MIS and MIS/SWCNTs. Obvious, after SWCNTs introducing, the redox peaks of MIS/SWCNTs at the first three cycle are almost overlapping, which indicated MIS/SWCNTs demonstrated a highly reversibility. Secondly, from Fig. 7f, it depicts the corresponding galvanostatic discharge-charge curves of the as-synthesize MIS/SWCNTs composites at different cycles (long term cycle). The results showed that during long-term cycles, the MIS modified by SWCNTs exhibited substantial overlapping behavior. The results showed that MIS/SWCNTs exhibit excellent reversibility, and also illustrated that the electronic structure after long cycles of charge and discharge is not significantly different from the initial three cycles.
Fig. 6 CV curves of (a) MIS and (b) MIS/SWCNTs at a scan rate of 0.1 mV/s.
Fig. 7(f) The charge-discharge profiles with different cycles for 100 cycles life test.

Reviewer 2 Report
Comments and Suggestions for Authors
In this study, a composite of transition metal sulfide and SWCNT was used as a negative electrode material for a lithium-ion battery, and it is believed to have shown good results. It would be good to find three areas of improvement.
1. The morphology of CNTs to prevent volume change cannot be accurately determined from SEM or TEM images. Please add a higher magnification image.
2. It was said that better conductivity was achieved through CNT, but it would be good to add the electrical conductivity of the composite itself.
3. Additionally, it would be good to supplement the scheme a bit more.
Comments on the Quality of English LanguageIt's somewhat well written.
Author Response
Reviewer 2
In this study, a composite of transition metal sulfide and SWCNT was used as a negative electrode material for a lithium-ion battery, and it is believed to have shown good results. It would be good to find three areas of improvement.
- The morphology of CNTs to prevent volume change cannot be accurately determined from SEM or TEM images. Please add a higher magnification image.
Response:
Thank you for reviewer’s good suggestion. We have provided a higher magnification image (HRTEM) of CNTs in revised Fig. S2.
Revised Fig. S2 HRTEM image of SWCNTs
- It was said that better conductivity was achieved through CNT, but it would be good to add the electrical conductivity of the composite itself.
Response:
Thanks to the reviewer's suggestion. We have conducted a four-point probe tests displayed in Fig. 1. It can be observed that the electronic conductivity of MIS modified with SWCNTs (24402.07 S/cm) was significantly improved compared to pristine MIS (110.35 S/cm), and the sheet resistance performance of MIS/SWCNTs (0.0137Ω) was also lower than that of pristine MIS (3.04 Ω).
Fig. 1(a) Conductivity and (b) sheet resistance of MIS and MIS/SWCNTs
- Additionally, it would be good to supplement the scheme a bit more.
Response:
Thank you for the reviewer’s comment. We have corrected Scheme 1 in the revised manuscript.
Revised Scheme 1.

Reviewer 3 Report
Comments and Suggestions for Authors
This manuscript proposed MnIn2S4/SWCNTs composite as anode materials for lithium ion batteries. The introduction of SWCNTs is attributed to suppress the dramatic volume changes during electrochemical reactions and it also improves the structural stability and electronic conductivity. The composite exhibited better cycling performances and improved capacity. Authors appropriately delivered obtained experimental results. Therefore, this manuscript is acceptable after major revision of following comments.
(1) As authors mentioned in the introduction part, there are many transition metal sulfide candidates as anode materials for lithium ion batteries. However, there are lacks of reason why authors chosen indium as the elements for the materials.
(2) Please specify the origin of the SWCNTs.
(3) What if more than 1 wt% of SWCNTs are added in the composites?
(4) Contribution of SWCNTs itself should be compared because SWCNTs also has its own ion storage behaviors.
(5) Please specify the constant voltage condition during discharge process in the experimental section. It seems the constant voltage capacity is quite contribute the whole capacity and cycling stability. It is not sure that the materials showed good value without the constant voltage capacity.
(6) MIS/SWCNTs also exhibited drastic capacity fading during initial stages of cycling tests (It seems lose almost ~200 mAh/g of reversible capacity). Even though this value is suppressed compared with MIS, the result is not considered as stable material as anode materials.
(7) Ex-situ XPS should be conducted to make sure the redox reactions of the transition metal including Mn and In.
Author Response
Reviewer 3
This manuscript proposed MnIn2S4/SWCNTs composite as anode materials for lithium ion batteries. The introduction of SWCNTs is attributed to suppress the dramatic volume changes during electrochemical reactions and it also improves the structural stability and electronic conductivity. The composite exhibited better cycling performances and improved capacity. Authors appropriately delivered obtained experimental results. Therefore, this manuscript is acceptable after major revision of following comments.
- As authors mentioned in the introduction part, there are many transition metal sulfide candidates as anode materials for lithium ion batteries. However, there are lacks of reason why authors chosen indium as the elements for the materials.
Response:
Thanks to the reviewer’s comments. We use indium as the anode material for lithium-ion batteries because of its potential advantages. First, indium has the characteristic of high lithium storage capacity, which means that it can store a large amount of lithium ions during the battery charging process. This can also increase energy density and extend battery life. Moreover, Indium-based anodes may experience less volume expansion and contraction during charging and discharging cycles compared to other materials. This reduced expansion can help mitigate mechanical stress on the battery structure, leading to improved stability and safety. This also gives it good rate capability and cycle life, which is also reflected in its rate capability and cycle life. And as you can see, the following published work all focus on indium-based anode materials for Li ion batteries.
- Sun, L.; Liu, X.; Ma, T.; L. Zheng, Y. Xu, X. Guo, J. Zhang, In2S3 nanosheets anchored on N-doped carbon fibers for improved lithium storage performances. Solid State Ionics 2019, 329, 8-14.
- Lee, T.Y.; Liu, W.R. Reduced graphene oxide-wrapped novel CoIn2S4 spinel composite anode materials for Li-ion batteries. Nanomaterials 2022, 12, 4367.
- Hsu, T.H.; Muruganantham, R.; Liu, W.R. High-energy ball-milling for fabrication of CuIn2S4/C composite as an anode material for lithium-ion batteries. Ceramics International 2022, 48, 11561-11572.
- Zhang, Z.; Yi, Z.; Liu, L.; Yang, J.; Zhang, C.; Pan, X.; Chi, F. Spray-drying assisted hydrothermal synthesis of ZnIn2S4@GO as anode material for improved lithium ion batteries. International Journal of Electrochemical Science 2020, 15, 8797-8807.
- Please specify the origin of the SWCNTs.
Response:
The SWCNTs we used is provided from Sino Applied Technology. The corresponding information has also added in the revised manuscript.
- What if more than 1 wt% of SWCNTs are added in the composites?
Response:
The reason why we choose 1wt.% SWCNTs in MnIn2S4 is because large content of SWCNTs may exhibit a worse electrochemical performance. As shown in reference [A1], this paper composites SnO2 with Ni and CNTs, and found that 1 wt.% CNTs was the optimal ratio. Therefore, we used 1 wt. SWCNTs to modify transition metal sulfide MIS, and then applies it to the anode material of lithium ion batteries.
- Ambalkar, A.A.; Kawade, U.V.; Sethi, Y.A.; Kanade, S.C.; Kulkarni, M.V.; Adhyapak, P.V.; Kale, B.B. A nanostructured SnO2/Ni/CNT composite as an anode for Li ion batteries. RSC Adv. 2021, 11, 19531-19540.
Fig. (a) The rate performances at different current densities of all samples between 0.005 and 3 V. (b) The cycling performances of all samples at a current density of 400 mA g−1 for 210 cycles. (c) The coulombic efficiencies of all samples at a current density of 400 mA g−1 for 210 cycles in lithium-ion batteries.[A1]
- Contribution of SWCNTs itself should be compared because SWCNTs also has its own ion storage behaviors.
Response:
Thanks to the reviewer's suggestions. According to the following references [A2,A3,A4], the capacity contribution of SWCNTs are investigated. Zhou et al proposed Fe2O3/SWCNT membrane as anode for LIBs. SWCNTs demonstrated a specific capacity of 187 mAh/g after 90 cycles at a current density of 500 mA/g [A2]. The ALD of GaSx thin films on a SWCNT support yields a nano-composite material was published by Meng et.al. SWCNTs depicted a SWCNTs also exhibit a specific capacitance of approximately 200 mAh/g after 100 cycles at a current density of 120 mA/g [A3]. Noerochim et al. reported SWCNTs/SnO2 prepared by vacuum filtration of SWCNT/SnO2 hybrid material, which was synthesized by the polyol method. Approximately 150 mAh/g of specific capacity at large current density of 1500 mA/g was achieved. [A4]. Based on these references, the reversible capacity of SWCNTs in this study may 150~200 mAh/g. Because we only use 1 wt% SWCNTs, thus, the capacity contribution of SWCNTs was only 1.5~2 mAh/g. Thus, we could neglect the contribution of SWCNTs. We hope reviewer could satisfy with our response.
- Zhou, G.; Wang, D.-W.; Hou, P.-X.; Li, W.; Li, N.; Liu, C.; Li, F.; Cheng, H.-M. A nanosized Fe2O3 decorated single-walled carbon nanotube membrane as a high-performance flexible anode for lithium ion batteries. Journal of Materials Chemistry 2012, 22, 17942-17946.
Fig. (a) Discharge–charge capacity versus current density plot of the Fe2O3/SWCNT and SWCNT membrane at different current densities. Solid symbols: discharge, hollow symbols: charge. (b) Cyclic performance of the Fe2O3/SWCNT and SWCNT membrane at a current density of 500 mA/g. Solid symbols: discharge, hollow symbols: charge.
- Meng, X.; He, K.; Su, D.; Zhang, X.; Sun, C.; Ren, Y.; Wang, H.H.; Weng, W.; Trahey, L.; Canlas, C.P. Gallium sulfide–single‐walled carbon nanotube composites: high‐performance anodes for lithium‐ion batteries. Advanced Functional Materials 2014, 24, 5435-5442.
Fig. (a) Cycling performance and columbic efficiency for commercial Ga2S3, commercial SWCNTs, and SWCNT-GaSx composite at 120 mA/g. (b) Cycling performance and columbic efficiency for the GaSx component of the SWCNT-GaSx composite at 120 mA/g.
- Noerochim, L.; Wang, J.-Z.; Chou, S.-L.; Wexler, D.; Liu, H.-K. Free-standing single-walled carbon nanotube/SnO2 anode paper for flexible lithium-ion batteries. Carbon 2012, 50, 1289-1297.
Fig. (a) Cycling stability of SWCNTs and SWCNT/SnO2 anode paper at constant current density of 25 mA/g; (b) high rate capability of the SWCNT/SnO2 anode paper
- Please specify the constant voltage condition during discharge process in the experimental section. It seems the constant voltage capacity is quite contribute the whole capacity and cycling stability. It is not sure that the materials showed good value without the constant voltage capacity.
Response:
Thanks to the reviewer's good suggestion. Our coin cell uses constant current and constant voltage to discharge in the voltage range of 0.01~3V. The terminate condition of constant voltage is that when discharging to the cut-off voltage, the current density will continue to gradually decrease to 1/10 of the original applied current.
- MIS/SWCNTs also exhibited drastic capacity fading during initial stages of cycling tests (It seems lose almost ~200 mAh/g of reversible capacity). Even though this value is suppressed compared with MIS, the result is not considered as stable material as anode materials.
Response:
Thanks to the reviewer's comments. When we conduct a 100-cycle cycle life test, the first three cycles were carried out at a smaller current density (0.1 A/g), then use a larger current density (0.2 A/g) to perform subsequent cycle charge and discharge. The “fading” was due to higher current density, not structural fading.
(7) Ex-situ XPS should be conducted to make sure the redox reactions of the transition metal including Mn and In.
Response:
Thank you for reviewer’s suggestions. In order to make sure the redox reactions of the transition metal including Mn and In, ex-situ XPS is indeed a very powerful tool. However, we can also use in-situ XRD measurement to identify phase evolution of MnIn2S4 during charge and discharge processes. Please see the following in-situ XRD data, which also support our proposed reaction mechanism.
Fig.1. (a) In-situ XRD patterns of MIS at 0.1C of first discharge/charge cycle, (b) The corresponding discharge /charge profile of MIS with identification of potential points of in-situ XRD
- Muruganantham, R.; Chen, J.A.; Yang, C.C.; Wu, P.J.; Wang, F.M.; Liu, W.R. Spinel phase MnIn2S4 enfolded with reduced gra-phene oxide as composite anode material for lithium-ion storage. Materials Today Sustainability 2023, 21, 100278.

Round 2
Reviewer 3 Report
Comments and Suggestions for Authors
As authors dealt with most of comments and questions appropriately, this manuscript is acceptable in this journal as presented form.